# A Genus-Wide Bioactivity Analysis of *Daboia* (Viperinae: Viperidae) Viper Venoms Reveals Widespread Variation in Haemotoxic Properties

**DOI:** 10.3390/ijms222413486

**Published:** 2021-12-15

**Authors:** Bianca op den Brouw, Francisco C. P. Coimbra, Nicholas R. Casewell, Syed Abid Ali, Freek J. Vonk, Bryan G. Fry

**Affiliations:** 1Venom Evolution Lab, School of Biological Sciences, The University of Queensland, St. Lucia 4072, Australia; francisco.cp.coimbra@gmail.com; 2Centre for Snakebite Research & Interventions, Liverpool School of Tropical Medicine, Liverpool L3 5QA, UK; Nicholas.Casewell@lstmed.ac.uk; 3Third World Center for Science and Technology, H.E.J. Research Institute of Chemistry, University of Karachi, Karachi 75270, Pakistan; dr.syedabidali@gmail.com; 4Naturalis Biodiversity Center, 2333 CR Leiden, The Netherlands; freek.vonk@naturalis.nl; 5Division of BioAnalytical Chemistry, Amsterdam Institute of Molecular and Life Sciences, Vrije Universiteit Amsterdam, De Boelelaan 1085, 1081 HV Amsterdam, The Netherlands

**Keywords:** *Daboia*, Russell’s viper, venom, toxin, venom variation, haemotoxicity, coagulopathy, fibrinogenolysis, snakebite envenoming

## Abstract

The snake genus *Daboia* (Viperidae: Viperinae; Oppel, 1811) contains five species: *D. deserti*, *D. mauritanica*, and *D. palaestinae*, found in Afro-Arabia, and the Russell’s vipers *D. russelii* and *D. siamensis*, found in Asia. Russell’s vipers are responsible for a major proportion of the medically important snakebites that occur in the regions they inhabit, and their venoms are notorious for their coagulopathic effects. While widely documented, the extent of venom variation within the Russell’s vipers is poorly characterised, as is the venom activity of other species within the genus. In this study we investigated variation in the haemotoxic activity of *Daboia* using twelve venoms from all five species, including multiple variants of *D. russelii, D. siamensis*, and *D. palaestinae*. We tested the venoms on human plasma using thromboelastography, dose-response coagulometry analyses, and calibrated automated thrombography, and on human fibrinogen by thromboelastography and fibrinogen gels. We assessed activation of blood factors X and prothrombin by the venoms using fluorometry. Variation in venom activity was evident in all experiments. The Asian species *D. russelii* and *D. siamensis* and the African species *D. mauritanica* possessed procoagulant venom, while *D. deserti* and *D. palaestinae* were net-anticoagulant. Of the Russell’s vipers, the venom of *D. siamensis* from Myanmar was most toxic and *D. russelli* of Sri Lanka the least. Activation of both factor X and prothrombin was evident by all venoms, though at differential levels. Fibrinogenolytic activity varied extensively throughout the genus and followed no phylogenetic trends. This venom variability underpins one of the many challenges facing treatment of *Daboia* snakebite envenoming. Comprehensive analyses of available antivenoms in neutralising these variable venom activities are therefore of utmost importance.

## 1. Introduction

The animal kingdom possesses a stunning diversity of venomous animals, with almost every major lineage containing a multitude of venomous species [1]. Of these, snakes are amongst the most infamous. Their notoriety stems from the susceptibility of humans to their venoms, which snakes may deploy when under perceived threat. Though sometimes used defensively, snake venom primarily functions to facilitate predation. The bioactive toxins that compose venom typically act upon molecular targets within the envenomed animal to disrupts its physiological processes, inducing weakness, immobility, or death [2]. Many toxins target molecules that are conserved within and between major animal groups, such as neuroreceptors or blood proteins [3,4]. Consequently, their activity is often effective in the bodies of a broad range of animals. This also partly explains the vulnerability of human physiology to many snake species’ venoms, particularly those that are adapted to feeding on mammals. 

Although a relatively small proportion of snake species are capable of causing serious injury or death in humans, their public health impact is severe [5]. Several medically important species can be found within the genus *Daboia* (Viperidae: Viperinae; Oppel, 1811), including the Russell’s vipers (*D. russelii* and *D. siamensis*)—notorious for their potent procoagulant venom and significant contribution to snakebite death in Asia. There are five species in the genus, which are geographically and phylogenetically split into two clades: the Afro-Arabian clade, which contains *D. deserti* and *D. mauritanica*, found in fragmented populations toward the far north of Africa, and *D. palaestinae*, found in parts of the Middle East; and the Asian clade, containing the Russell’s vipers *D. russelii*, found extensively throughout eastern Pakistan, India, Sri Lanka, and Bangladesh, and *D. siamensis*, found in disjunct pockets throughout Southeast Asia. Envenomings by all five species have the potential to cause mortality. 

Disruption of haemostasis is a ubiquitous and often fatal outcome of *D. russelii* and *D. siamensis* envenomings [6,7,8]. Despite this shared pathophysiology, Russell’s viper venoms differ extensively between populations, manifesting distinct additional pathologies that respond variably to antivenoms and impact upon therapeutic success [9,10,11]. While the need for venom characterisation and preclinical assessment of antivenoms is being increasingly addressed for Indian *D. russelii* (e.g., [7,12,13,14,15,16], there are presently only a handful of studies that characterise the venoms of *D. russelii* from Pakistan and Sri Lanka or *D. siamensis* from Myanmar, Taiwan, Thailand [7,15,17,18,19,20,21,22,23,24,25,26,27], and even fewer that describe the venom of Indonesian populations [28,29]. 

The venom profiles of the Afro-Arabian clade have received even less research attention, presumably on account of their considerably lower contribution to the global snakebite burden [30]. However, the comparatively few human fatalities by these species is more reflective of lower population densities with less overlap between humans and snakes than it is indicative of low venom toxicity in humans. Proportionally, the Afro-Arabian *Daboia* are the most medically important species in their respective regions [30,31,32,33,34]. Despite this, studies on their venoms are scarce. *Daboia deserti* venom is mostly undescribed. The few studies characterising the venom of *D. mauritanica* document snake venom metalloproteases (SVMPs), haemorrhagins, and fibrinogenolytic activity [31,35,36,37], while *Daboia palaestinae* venom is said to contain haemorrhagic SVMPs, neurotoxic phospholipases (PLA_2_s), and snake venom serine proteases (SVSPs) [33,34,38,39,40]. 

As haemostatic disturbances appear to be a dominant and critical outcome of envenoming by all *Daboia* species, bioactivities associated with haemotoxicity were the focal point of this study. Proteolytic activity of snake venom metalloproteases, serine proteases, and phospholipases was measured by fluorometry. Coagulometric analyses (intensity and potency of coagulant activity, clot strength, and clotting kinetics) of the venoms in human plasma were conducted using a Stago STA R MAX^®^ automated haemostasis analyser, Haemonetics TEG^®^ 5000 thromboelastography, and Stago Calibrated Automated Thrombography (CAT). Activation of human factor X and prothrombin by the venoms was measured by fluorometry using a Fluoroskan Ascent™ Microplate Fluorometer. Activity on fibrinogen was assessed using TEG^®^ 5000 thromboelastography and by fibrinogen gels. For all analyses we used twelve distinct venom samples sourced from all five species of *Daboia* to provide the first genus-wide characterisation of *Daboia* venom activity.

## 2. Results and Discussion

### 2.1. Protease Activity

#### 2.1.1. Venom Serine Protease and Metalloprotease Activity 

There was substantial variation between the venoms in their activity on fluorogenic substrates ES001 (for matrix metalloproteases collagenase 1–3, gelatinase A-B), ES002 (matrix metalloproteases stromilysin 1–2 and serine proteases trypsin and factor Xa;), and ES011 (serine proteases factor II and VII, kallikrein) (Figure 1). Of the Asian group, only *D. russelii* from India and *D. siamensis* from Myanmar had activity on substrate ES001, while activity was present by all the venoms of the Afro-Arabian group except *D. mauritanica*. As this substrate is cleaved by collagenases and gelatinases, this may represent the action of haemorrhagic snake venom metalloproteases (SVMPs) on extracellular matrices. However, as these substrates are not highly specific, this venom activity cannot be confidently assigned to an in vivo function. For substrates ES002 and ES011, the Afro-Arabian group generally exhibited greater substrate cleavage than the Asian group. While associated function is again unclear, these differences can be used as a proxy for comparing relative activity of the SVMPs and snake venom serine proteases (SVSPs) constituting these venoms.

#### 2.1.2. Venom Phospholipase Activity

Relative enzymatic phospholipase A_2_ (PLA_2_) activity, measured via fluorescent kinetic assay, also varied between the *Daboia* venoms (Figure 2). Activity levels was similarly high in all species of the Asian group except for *D. siamensis* from Myanmar, which was substantially lower than the rest of the clade. Activity was low-moderate for the Afro-Arabian group. Consistent with proteomic analyses in the literature [7], the Sri Lankan population appeared to possess amongst the greatest PLA_2_ activity. For most venoms, PLA_2_ activity increased non-linearly relative to venom concentration increase, though again this varied. For example, a two-fold increase in venom concentration (2 to 4 ng/mL) elicited a 12.6-fold increase in PLA_2_ activity in *D. russelii* from Pakistan compared to a 2.5-fold increase in activity in *D. siamensis* from Java. These non-linear increases are suggestive of synergistic PLA_2_ activity, as has been described from the venom of east-Indian *D. russelii* [41]. Phospholipases are a significant component of *Daboia* venoms and are known to interact with a diversity of molecular targets and biochemical pathways, including those involved in blood, tissue, and nerve regulation [41,42,43,44]. Further investigation of PLA_2_ function in these venoms was beyond the scope of this study, though presents an important area of future research.

### 2.2. Coagulant Activity

As haemostatic disturbances are a critical feature of *Daboia* envenoming, this characteristic was the focal point of all further assays in this study. The venoms were tested for their procoagulant activity on citrated (3.2%) human plasma, both with and without co-factor calcium. Strength of the venoms (i.e., intensity of their procoagulant activity) was evaluated by saturating plasma with venom and measuring time until clot formation (minimum clot time). Venom potency (concentration required to induce clotting of a given intensity) was assessed by conducting 9-point dose-response curves (0.05–30 µg/mL). From these, mid-curve coagulation times along with corresponding concentrations were obtained. The curve data were also used to statistically calculate EC_50_s (50% of the maximal Effective Concentration), which denote the venom concentration required to induce the clot time at the mid-point between maximal strength of activity (minimum clot time) and no activity (vehicle control clot time; ~422 s). 

#### 2.2.1. Intensity of Coagulant Activity

In the absence of calcium, the venoms were unable to induce coagulation (data not shown). All venoms from the Asian clade attained their minimum clot times with 20 µg/mL venom and induced clot formation in recalcified plasma in less than 14 s (Figure 3A). The minimum clot times by the populations of *D. russelii* differed marginally (by ≤ 3 s) though significantly (*p* < 0.0001), with the Sri Lankan venom being slowest to induce a clot (13.7 s; 95% CI: 13.3–14.1 s) and the Indian venom the fastest (10.4 s; 95% CI: 10.0–10.9 s) (see Appendix A). All *D. siamensis* venoms produced a similar minimum clot time of around 10 s (*p* = 0.0563), and all were faster than any of the *D. russelii* venoms.

As minimum clot times were established by saturating the plasma with venom—a biologically unrealistic scenario in relation to human snakebite envenoming—mid-curve coagulation times were compared to assess retention of procoagulant activity with weakening venom concentrations. Of the *D. russelii* venoms, the Indian sample demonstrated the greatest retention of activity, with clotting time slowing by 1.8-fold. The Sri Lankan *D. russelii* venom’s mid-curve clot time of 29.1 s (95% CI: 21.1–37.1 s) represented a 2.1-fold shift and was again significantly slower than its conspecifics (*p* ≤ 0.0378), though clotting by the Pakistani *D. russelii* venom slowed by a slightly greater proportion (2.2-fold). All the *D. siamensis* venoms demonstrated high retention of activity, with mid-curve clotting times between 17.9–19.3 s. Despite this narrow range the difference was significant (*p* = 0.0011), which is notable due to the lack of significant difference between their minimum clot times. Clotting by the Thai and Javan venoms slowed by the greatest proportion (1.9-fold), followed by that from Taiwan (1.8-fold) and Myanmar (1.7-fold). 

The venoms of the Afro-Arabian group promoted clotting slowly and with broad variability between venoms (*p* < 0.0001). *Daboia mauritanica* reached its maximal procoagulant activity with 30 µg/mL venom, which induced a clot in 58.4 s (95% CI: 48.7–68.1 s). *Daboia deserti* and *D. palaestinae* (Israel 1) venoms also achieved their maximal activity with 30 µg/mL venom, though their mean minimum clot times of > 150 s represents very weak procoagulant activity. In contrast to their conspecific, the other two *D. palaestinae* venoms did not appear to promote coagulation, producing minimum clotting times comparable to that of the plasma’s spontaneous clotting time (*p* ≥ 0.1039). 

While the minimum clot time of *D. palaestinae* (Israel 2) was produced by 30 µg/mL venom, that of the *D. palaestinae* venom of unknown location occurred with 0.05 µg/mL. Furthermore, higher concentrations of the latter venom slowed clotting, with 20 µg/mL venom delaying clot formation by a further minute and 30 µg/mL by approximately two minutes or more (see Appendix A). While clotting did still generally occur, this reduction in clotting speed with higher concentrations is suggestive of an anticoagulant mechanism within the venom (see TEG and fibrinogenolytic assay sections for further discussion). The venom of “Israel 2” demonstrated variable coagulations times with limited consistency between replicates or trends related to concentration, which may reflect indiscriminatory or competitive proteolytic activity. 

There are no further details pertaining to the three *D. palaestinae* venoms, including age or precise location of the sampled snakes, both of which often correlate with intraspecific venom variation when also associated with temporal or geographical differences in diet [45,46,47]. As *D. palaestinae* shifts from a diet of lizards to rodents and birds with age [48], the considerable differences in activity between these *D. palaestinae* venoms could reflect ontogenetic venom variation, though further studies are required to test this hypothesis.

#### 2.2.2. Potency of Coagulant Activity

All venoms of the Asian clade demonstrated very high potency in plasma, with all EC_50_s being less than 0.3 ng/mL (Figure 3B). Of the *D. siamensis* venoms, which ranged between 0.05–0.07 ng/mL, those from Myanmar and Taiwan were equal-most potent and the Thai population the least, though these differences were insignificant (*p* = 0.6815). Of the *D. russelii*, the Indian locality was the most potent with an EC_50_ of 0.05 ng/mL (95% CI: 0.02–0.09 ng/mL), while the Sri Lankan venom was the least potent by a considerable margin (0.30 ng/mL, 95% CI: 0.14–0.57 ng/mL). Accordingly, the EC_50_s of the *D. russelii* venoms differed significantly (*p* = 0.0039). 

Due to the very high potency of these venoms and sensitivity of the assay, 95% confidence intervals of the EC_50_s were wide. Therefore, the venom concentrations that corresponded to the mid-curve clotting times produced by the concentration range tested in the dose-response curves (i.e., between 0.05–30 μg/mL) were also compared as a supporting analysis (Figure 3D). For the Asian clade, all mid-curve concentrations were lower than 0.60 µg/mL and followed similar trends as those evident in the EC_50_s. The *D. russelii* venoms differed significantly (*p* < 0.0001), ranging between 0.38–0.52 μg/mL, and the least potent venom again belonged to the Sri Lankan locality. No significant difference between the *D. siamensis* venoms was observed (*p* = 0.0546), all of which ranged between 0.34–0.37 µg/mL. This lack of significant difference is notable as these *D. siamensis* mid-curve concentrations generated significantly different mid-curve clotting times (Figure 3C), despite no difference between their minimum clot times (~10 s) and associated concentrations (20 μg/mL). Consequently, the variability between the *D. siamensis* mid-curve clotting times may reflect differing affinities of the procoagulant toxins in their venoms.

The venoms of the Afro-Arabian clade were again substantially weaker and ranged widely, with EC_50_s between 285–4531 ng/mL. While *D. mauritanica* had a relatively low mid-curve concentration (0.80 µg/mL, 95% CI: 0.74–0.87 μg/mL), the corresponding clot time was slow (183.7 s, 95% CI: 179.4–188 s). Mid-curve values for *D. palaestinae* and *D. deserti* further confirm that both venoms have low potency (mid-curve concentrations > 2.6 µg/mL) and intensity of coagulant activity (mid-curve clot times > 270 s). 

#### 2.2.3. Clot Kinetics 

Following the coagulation time assays, thromboelastograms were run to assess clotting kinetics and the impact of the venoms on clot polymerisation. Split point (SP), the time at which clot formation begins, signifies the initiation period of enzymatic activity. The venoms of the Asian clade showed rapid onset, with clot formation commencing in less than 0.40 min (Figure 4). This exceeded the rate of the FXa and thrombin controls–by as much as two to three times, in the case of the D. siamensis from Myanmar. All *D. siamensis* venoms were significantly faster than the FXa control (*p* ≤ 0.0286). As with the coagulometric assays, all the *D. siamensis* venoms were faster than the *D. russelii* venoms, the latter of which took at least 0.35 min to initiate clot formation. Within species, differences between venoms were evident though not statistically significant (*D. russelii: p* = 0.7538; *D. siamensis: p* = 0.9584). 

The reaction time (R) denotes the time by which the clot has reached an amplitude of 2 mm. This, when viewed in conjunction with the split point, gives indication as to the conversion rate of prothrombin into thrombin upon clot onset, whereby a greater difference between SP and R signifies a slower burst rate and lower rate of enzyme activity. Based on these parameters, *D. siamensis* venoms from Myanmar and Taiwan induced the greatest rate of prothrombin conversion. Notably, the time between SP and R by these venoms was indiscernible and thus clot formation from onset to 2 mm was virtually instantaneous, indicating an incredibly rapid rate of enzyme activity. 

In addition to expediting the clot initiation period, all venom of the Asian clade more than doubled the rate of clot polymerisation, which reflects the interactions of the toxins and activated factors with fibrinogen, compared with the rate of spontaneous clot polymerisation, which formed according to endogenous concentrations of blood factors within the plasma. Furthermore, all the Asian venoms exceeded polymerisation rates by both the FXa- and thrombin-induced clots (Figure 5). The clots produced by the venoms were of comparable strength to those of the controls (*p* = 0.7352). 

These data demonstrate that the procoagulant activity of these venoms stimulates rapid, strong clots via the swift generation of thrombin and fibrin. The venoms consistently matched or exceeded the positive controls, which likely reflects concurrent activation of multiple coagulation factors with high efficacy.

Most venoms of the Afro-Arabian clade impeded clot formation and strength. This was particularly pronounced for those of the two Israeli *D. palaestinae* venoms, by which clotting was completely or near-completely inhibited. While the *D. palaestinae* venom of unknown locality produced a clot, initiation was significantly delayed (*p* = 0.0006) and the resulting clot’s structure was significantly weakened compared to the vehicle control spontaneous clot (*p* = 0.0084). These data indicate that all the *D. palaestinae* venoms are anticoagulant in human plasma. While the *D. deserti* venom demonstrated no significant effect on clot initiation time (*p* = 0.3656), the venom produced significant reductions in clot polymerisation rate (*p* = 0.0004) and strength (*p* = 0.0007) compared to the vehicle control. This indicates that, while some clotting occurs, the venom of *D. deserti* interferes with blood proteins to ultimately hinder clot formation and stability and should also be classed as an anticoagulant venom in human plasma. In contrast, the *D. mauritanica* venom expedited clot initiation (*p* = 0.000046) and polymerisation rate (*p* = 0.00031) compared to the vehicle control and produced a clot that was as structurally firm as those of thrombin and FXa (*p* ≥ 0.3161). Although the *D. mauritanica* venom was more than five times slower to initiate clotting than the incredibly rapid Asian venoms, its fast rate of clot formation and the structural integrity of the resulting clot confirms that this species possesses a moderately strong procoagulant venom in human plasma.

This contrast in coagulant activity between the *D. mauritanica* and *D. deserti* venoms is notable as recent genetic analyses have questioned the validity of their taxonomic separation and suggested that *D. deserti* be reclassed as a subspecies of *D. mauritanica* [49]. The exact localities of the venom samples tested in this study are not known, which poses questions surrounding the venom composition of *D. mauritanica* across its distribution and the influence of vicariance events on the venom of the *D. mauritanica-deserti* lineages. These taxa therefore offer an interesting model to study venom evolution in future studies. Crucially, this venom variation is an important finding in relation to snakebite therapeutics for this region and raises concerns regarding future production and marketing of antivenoms for these snakes, should the reclassification become widely applied. 

#### 2.2.4. Thrombin Generation

To further dissect the differences in coagulant activity, real-time thrombin generation by the venoms was quantified using calibrated automated thrombography (CAT). There were considerable differences in peak height (maximum concentration of thrombin generated) and time to peak between the venoms of *D. siamensis* (Figure 6A). However, those with a longer initiation phase had greater peak heights, which resulted in high endogenous thrombin potential (ETP) across all samples (between 1673–1865 nmol/L x min; see Appendix A). Although *D. russelii* from Sri Lanka had a substantially lower peak concentration and time to peak than its conspecifics, ETP values between *D. russelii* venoms were also similar (between 1413–1467 nmol/L × min) (Figure 6B). Consequently, total thrombin generation trends within the Asian clade were broadly congruent with previous assays whereby the *D. siamensis* venoms consistently achieved ETP values that exceeded those of the *D. russelii* venoms. Beyond this, there was considerable disparity between the thrombin generation and the venom-induced coagulation results. Notably, the *D. siamensis* venoms from Thailand and Myanmar had substantially longer lag times despite being amongst the fastest at inducing clotting, and the venoms of the Afro-Arabian clade (Figure 6C,D) generated values that were comparable to those of the *D. russelii* populations. However, CAT measures thrombin generation as opposed to clot formation, and these discrepancies may be the result of differing concentrations of anticoagulant or fibrinogenolytic toxins between venoms that variably impact upon clot times. 

#### 2.2.5. Fibrinogenolytic Activity

To determine the activity of the venoms on fibrinogen and subsequent impact upon clotting, fibrinogen gels and fibrinogen-only thromboelastograms were conducted. If procoagulant fibrinogenolysis by a venom was not evident (i.e., no clot formed during the 30-min thromboelastography run), anticoagulant fibrinogenolysis was assessed by adding thrombin to the mixture and the run was continued for an additional 30 min. The thrombin induced clotting of any fibrinogen that had remained functionally unaffected by the venoms, and this was used to quantify venom-mediated fibrinogen degradation. 

Notably, only the venom of *D. siamensis* from Myanmar demonstrated procoagulant fibrinogenolysis (Figure 7A), and its action appeared restricted to the Aα-chain (Figure 7A(I)). Both fibrinopeptides A (FpA), of the Aα-chain, and B (FpB), of the Bß-chain, must be cleaved by thrombin in order to produce fully functional fibrin polymer strands. The cleavage of FpA alone results in fibrin monomers that produce a weakened, friable clot [50,51], which is reflected by the reduced strength of the Myanmar venom-induced clot compared to that of thrombin (*p* = 0.046727). However, in a system complete with prothrombin, the venom simultaneously generates the thrombin required to accomplish FpB cleavage and produce strong, polymerised clots. This is likely a synergistic system that facilitates endogenous thrombin efficiency, and may explain the disparity between the venom’s rapidity of clot formation seen in the plasma assays and the extended lag time seen in the thrombin generation assay, in which the contribution to clotting by direct action on fibrinogen is not quantified.

All other venoms of the Asian clade possessed anticoagulant fibrinogenolytic activity, along with all members of the Afro-Arabian clade. The diverse patterns of fibrinogen chain cleavage (Figure 7A(I),B(XI)) and differing intensities of fibrinogen degradation (Figure 7B) by the venoms appeared to be unlinked to phylogenetic relationships. The Asian venoms reduced functional fibrinogen by between 11–95%, while the venoms of the Afro-Arabian clade similarly ranged between 14–99%.

A notable result was the destructive cleavage of all three fibrinogen chains by the Indian *D. russelii* venom (Figure 7B(IV)), rendering virtually all the fibrinogen incoagulable. While this may reflect the specific action of fibrinogenolytic toxins with high affinity, potency, and/or concentration, the complete and apparently non-preferential lysis of all three chains near-simultaneously suggests that this may be the epiphenomenal action of indiscriminate proteases [52]. Similar, though weaker, degradation of all three fibrinogen chains was observed in the Javan *D. siamensis* venom (Figure 7B(I)). The strength of anticoagulant fibrinogenolysis by these two “true procoagulant” Russel’s viper venoms was comparable to the “true anticoagulant” venoms of the Afro-Arabian clade, in contrast to the negligible reduction of fibrinogen by the venom of *D. russelii* from Sri Lanka (*p* = 0.4204).

The fibrinogenolytic activities of the Afro-Arabian clade venoms also differed widely. While the venom of *D. palaestinae* from Israel (sample 2) near-completely degraded the fibrinogen (99 ± 0.4% reduction), its conspecifics reduced functional fibrinogen by around half that amount. Furthermore, the preferential cleavage of the Bβ-chain by the “unknown” *D. palaestinae* venom was not shared by the venoms of its conspecifics (Figure 7B(VII–IX)), though was demonstrated by that of *D. deserti* (Figure 7B(X)). Confounding results were presented by the *D. mauritanica* venom (Figure 7B(XI)), whereby the Aα and Bß chains were cleaved yet virtually all fibrinogen appeared to remain coagulable, as no significant difference to the control clot size was evident (*p* = 0.099519). This suggests that the venom-induced cleavage was neither clot-promoting nor clot-inhibiting, the mechanisms of which are elusive.

The diversity of actions on fibrinogen evident throughout these data indicates that fibrinogenolytic activity of *Daboia* venoms is a dynamic character. Given that any similarities in fibrinogenolysis between venoms are punctuated throughout the genus, whether shared activities are the result of homologous toxins with differing rates of expression between venoms or are the action of distinct toxins that have converged upon a shared target is ambiguous. The lack of trends relative to phylogeny suggests that these toxins may serve an ancillary role that is not subject to strong selection pressure, and these widely differing activities may reflect genetic drift. 

Regardless of evolutionary origin, the fibrinogenolysis exerted by these venoms is likely of sufficient capacity to contribute to the defibrinogenation experienced by envenomed patients. Furthermore, fibrinogen, which is typically present between 2–4 mg/mL in human plasma, is an acute phase protein whose concentration can increase to 7 mg/mL in response to acute inflammation [53]. Inflammation is commonly associated with envenoming by viperids, including *Daboia* [54], and the contribution of these fibrinogenolytic toxins to venom pathogenesis may be linked to the venom-induced inflammatory response. 

#### 2.2.6. Factor X and Prothrombin Cleavage

To identify the haemostatic proteins being activated by the venoms and to elucidate the unexpectedly high levels of thrombin generated by the Afro-Arabian clade, the venoms were tested for their ability to cleave blood proteins factor X (FX) and prothrombin using human recombinant factors in fluorescence assays. 

All species cleaved both factor X and, to a substantially lesser extent, prothrombin (Figure 8). The venoms of the Asian clade possessed very high total FX cleavage activity (area under the curve; fluorescence × time), achieving between 77–89% of that produced by the FXa positive control. The venom of *D. siamensis* from Myanmar was the greatest of these, while the venom of the Indian *D. russelii* was the lowest. The maximum rates of reaction by the Asian venoms were, however, only between 30–60% that of FXa, and their lag times were at least five-fold longer. The Indian *D. russelii* venom again trailed behind all other members of the Asian clade, attaining the slowest rate and a lag time twelve-fold longer than that of FXa. The *D. siamensis* venoms did not significantly differ in their lag times (*p* = 0.7131), though differences between their peak enzymatic rates (*p* = 0.001) produced significantly differing total activity levels (*p* = 0.0004). The FX activation values produced by the *D. russelii* venoms varied significantly across all parameters measured (*p* ≤ 0.0145) and were consistently weaker than those of the *D. siamensis*.

Prothrombin cleavage was evident throughout the Asian clade at similar levels among venoms, though their activity was very weak and extremely slow, with total activity at only 9–12% of the thrombin control, peak rates between 13–18%, and lag times typically exceeding 30 min compared to the ~1 s by thrombin. Despite these relatively narrow ranges, the differences between geographical variants within each species were significant for all parameters (*p* ≤ 0.0014), except for cleavage rate by the *D. siamensis* venoms (*p* = 0.3200). Notably, in contrast to factor X activation, the greatest total activity and fastest lag time on thrombin by these venoms was achieved by the Indian *D. russelii*. 

The venoms of the Afro-Arabian clade cleaved FX at substantially reduced levels compared with the FXa control and the Asian clade, attaining maximum cleavage rates of between 5–12% that of the FXa control, and taking at least 17 times longer to do so. Total activity by these venoms was low to moderate, ranging from 23–61% of the FXa control. The FX cleavage activity by the *D. palaestinae* “unknown” sample consistently exceeded all venoms of the clade, while that by *D. mauritanica* was consistently weakest. Activity by the *D. palaestinae* venom variants differed significantly across each parameter (*p* ≤ 0.0192). 

Notably, prothrombin cleavage activity was greater within this clade than the Asian clade. Although they were also very slow to initiate, Afro-Arabian venoms achieved total cleavage activities of 54–59% and rates between 26–28% of the prothrombin control. This may account for the unexpectedly high total thrombin generation by these venoms during the thrombograms along with the weak coagulant activity observed in the plasma coagulation assays, which corroborates with a number of case reports indicating mild coagulopathy and haemostatic disturbances following envenomings by these species [30,33,34,37,55].

The posited activation of factor X and prothrombin by *D. mauritanica* venom either differed little or was weaker than that by the *D. deserti* and *D. palaestinae* venoms, despite this venom demonstrating significantly stronger procoagulant activity. A similar discrepancy between strength of plasma clotting activity and relative activation of factor X or prothrombin has also been noted for some venoms of the closely related genus *Macrovipera*, which activate factor V as well as factor X [56,57,58]. Factor V activators have been previously documented from venoms of the Asian *Daboia* species [19,56,57], and it is possible that the presence of such toxins in the *D. mauritanica* venom may underpin the disparities between relative factor activation and coagulant response within the Afro-Arabian clade. However, toxins that are inducing a parallel anticoagulant response and slowing coagulation time in the plasma assays may also be confounding the results for the *D. deserti* and *D. palaestinae* venoms. 

The venoms of Russell’s vipers are infamous for their factor X activating toxins (e.g., [58,59,60]) and these data support the notion that such toxins are likely present in the venoms of most—if not all—*Daboia* populations. Prothrombin activators have previously been described from *Daboia* venoms, though on very few occasions [12,61,62,63]. Although the prothrombin cleavage activity observed in this study was substantially weaker than that for factor X, these results allude to the possibility that prothrombin activating toxins may also be more widespread amongst *Daboia* than previously presumed, albeit in very low concentrations, and could be conferring an ancillary contribution to the coagulopathy observed in envenomed patients. It is however important to recognise that these data do not unequivocally demonstrate functional prothrombin as, depending on the nature of prothrombin cleavage, this activity may not necessarily confer the activation of prothrombin into a functional form of thrombin. However, the thrombin generation and clotting activity observed in the CAT and TEG assays support the conjecture.

The genus-wide presence of these putative factor X and prothrombin activators and the procoagulant activity of *D. mauritanica* venom provides modest evidence to suggest that procoagulant venom could be an ancestral trait within *Daboia*, which has been amplified in the common ancestor of the *D. russelii* and *D. siamensis* complex. 

## 3. Materials and Methods

### 3.1. Venom Samples

Venoms were obtained from multiple sources, detailed in Table 1. Venoms were immediately flash frozen in liquid nitrogen and were subsequently lyophilised and stored at −80 °C. Working stock aliquots used for all experiments consisted of lyophilised venom reconstituted to 1 mg/mL in a solution of 1:1 deionised water (dH_2_O) and glycerol (Sigma-Aldrich) to prevent freezing, and were stored at −20 °C. Experimental venom work was conducted under the University of Queensland IBSC approval #IBC134BSBS2015.

### 3.2. Plasma

Pooled human plasma (citrated 3.2%) was sourced from the Australian Red Cross (Kelvin Grove, QLD) under human ethics approval number 2016000256. Two batches (Lot#3379236: 120 mL of Rhesus O+; Lot#3376558: 874 mL of Rhesus AB+) were further combined and aliquoted, flash frozen in liquid nitrogen and stored at −80 °C. Immediately prior to use, aliquots were thawed in a water bath for 5 min (min) at 37 °C. As clotting times can vary between plasma batches, the coagulation parameters of this mixture were established by measuring clot formation time in the presence and absence of a clotting activator (kaolin): 46–48 s and ~300–500 s, respectively (see Section 3.4.1 for protocol). These were then used as reference values for the positive and negative controls conducted prior to each experiment over the course of the study to ensure consistency. 

### 3.3. Enzyme Activity Assays

Venom enzyme activity was assessed by combining venoms with fluorescent substrates using a Fluoroskan Ascent™ Microplate Fluorometer with Ascent^®^ Software v2.6 (Thermo Fisher Scientific, Vantaa, Finland). Unless otherwise stated, venom working stocks were diluted to the desired concentration in dilution buffer (150 mM NaCl, 50 mM Tris-HCl, pH 7.4), each condition was tested in triplicate using 384-well plates, and machine incubation temperature was set at 37 °C.

#### 3.3.1. Snake Venom Phospholipase A_2_ Activity (PLA_2_)

Venom phospholipase A_2_ (PLA_2_) activity was tested using the reagents and methods of EnzChek^®^ Phospholipase A_2_ Assay Kit (Thermo Fisher Scientific, Rochester, NY, USA). Briefly, 0.1 µg venom was brought up to 13 µL in reaction buffer (250 mM Tris-HCL, 500 mM NaCl, 5 mM CaCl_2_, pH 8.9) and loaded into a well. EnzChek^®^ Phospholipase A_2_ substrate was dispensed into each well (13 µL quenched 1 mM). Readings commenced immediately with fluorescence measured at excitation of 485 nm and emission of 520 nm and were continued for 100 cycles at room temperature. Negative control wells contained 13 µL PLA_2_ reaction buffer and positive control consisted of kit-provided purified PLA_2_ from bee venom (5 Units/mL) in 13 µL PLA_2_ reaction buffer.

#### 3.3.2. Snake Venom Metalloprotease (SVMP) and Serine Protease (SVSP) Activity

Fluorogenic Peptide Substrates Cat#ES001, Cat#ES002, and Cat#ES0011 (R&D systems, Minneapolis, MN, USA) were used to test for SVMP, SVMP/SVSP, and SVSP activities, respectively. Wells were loaded with 10 μL of venoms (0.01, 0.05, and 0.1 µg/μL) or 10 µL dilution buffer (negative control). Ninety microlitres of quenched substrate was automatically dispensed (1 µL substrate per 500 μL enzyme buffer: 150 mM NaCl, 50 mM Tris-HCl, 5 mM CaCl_2_, pH 7.4) and fluorescence was read at 390 nm (excitation) and 460 nm (emission) (Cat#ES001, Cat#ES002) or 320 nm (excitation) and 405 nm (emission) (Cat#ES0011) for 300 cycles over approximately 150 min. 

### 3.4. Venom-Induced Coagulant Activity 

#### 3.4.1. Coagulometric Assays Using STA-R Max^®^

Venom-induced clotting time analyses were performed using a Stago STA-R Max^®^ automated coagulation analyser and Stago Analyser software v0.00.04 (Stago, Asnières sur Seine, France). Maximal clotting activity (i.e., minimum clot time) of each venom was determined by increasing venom concentration until clot times no longer decreased. Venoms were then tested at nine concentrations in triplicate (final venom concentration, μg/mL: 30.00, 10.00, 4.00, 1.67, 0.67, 0.25, 0.13, and 0.05). Experimental conditions were as follows: venom working stocks were manually diluted to 0.1 mg/mL in Owren Koller (O.K.) Buffer (Stago Cat# 00360) and loaded into the analyser, after which all subsequent dilutions, steps, and measurements were automated. In a ~250 μL cuvette, 50 μL venom (automatically diluted in O.K. Buffer to achieve desired concentration) was incubated with 50 μL of calcium (CaCl_2_, 0.025 M, Stago Cat# 00367), 50 μL phospholipid (STA C.K. Prest, Stago Cat# 00597, solubilised in O.K. Buffer; unknown concentration), and 25 μL of O.K. Buffer for 120 s at 37 °C. Then, 75 μL of pre-incubated (37 °C) human plasma was added and subsequent clotting time was measured by the analyser using a viscosity-based detection system. To measure toxin co-factor dependency, calcium was substituted with 50 μL O.K. buffer and tested with 20 μg/mL venom. Negative control conditions replaced 50 μL of venom with vehicle solution (*v/v*: 5% glycerol, 5% H_2_O, 90% OK buffer). Positive control conditions were conducted by incubating 50 μL Kaolin (STA C.K. Prest standard kit, Stago Cat#00597) with 50 μL plasma for 120 s at 37 °C, following which 50 μL CaCl_2_ (0.025 M, Stago Cat#00367) was added and time until clot formation measured. 

#### 3.4.2. Thromboelastography on Plasma and Fibrinogen

The effects of venoms on clot dynamics of plasma and fibrinogen were measured using a Thromboelastograph^®^ 5000 Haemostasis analyser (Haemonetics^®^, Haemonetics Australia Pty Ltd., Sydney, Australia). In quadruplicate: 189 μL of human plasma or human fibrinogen (4 mg/mL in buffer (150 mM NaCl, 50 mM Tri-HCl, pH 7.3), Sigma Aldrich, Sydney) was combined with 72 μL CaCl_2_ (25 mM, Stago Cat# 00367 STA), 72 μL phospholipid (Stago Cat#00597 in O.K. buffer), and 20 μL O.K. buffer in an assay cup (Cat.# 07-052, Haemonetics, Sydney Australia) incubated at 37 °C. Following this, 7 μL venom working stock was added (final venom concentration: 20 μg/mL) and briefly pipette mixed, and the 30-min run was immediately initiated. If no clots were formed by the venoms after 30 min, to assess fibrinogen degradation by the venoms, 7 μL thrombin (STA^®^-Liquid Fib, Stago Cat#00673, Asniéres sur Seine, France; unknown concentration) was added to the cup to induce clotting of any remaining functional fibrinogen and the measurements were continued for another 30 min. Negative and positive controls substituted 7 μL venom with 7 μL vehicle solution or 7 μL thrombin (Liquid Fib^®^, Cat#115081, Stago; unknown conc.) or FXa (STA^®^-Liquid Anti-Xa, Stago; unknown conc.).

#### 3.4.3. Fibrinogen Gels

Fibrinogen gels were conducted using well-established protocols described in [64]. Briefly, a 50 μL aliquot of preincubated (37 °C) human fibrinogen (Sigma Aldrich, Sydney) (1 mg/mL in buffer (150 mM NaCl, 50 mM TrisHCl, 5 mM CaCl, pH 7.4)) was placed in a heat block at 37 °C. Ten microlitres was removed, combined with 10 μL loading and reducing buffer (5 μL of 4× Laemmli sample buffer (Bio-Rad, Hercules, CA, USA), 5 μL deionised H_2_O, 100 mM DTT (Sigma-Aldrich, St. Louis, MO, USA)), boiled at 100 °C for 4 min, and set aside (“0 min incubation” fibrinogen control). Then, 4 μL venom working stock (1 mg/mL) was added to the incubated aliquot containing 40 μL (final venom concentration: 0.1 mg/mL). At intervals of 1, 5, 20, and 60 min, a further 10 μL was removed from the incubated aliquot and combined with 10 μL loading and reducing buffer, boiled at 100 °C for 4 min, and set aside. These aliquots were then loaded into a 1 mm 12% SDS PAGE gel and were run in 1× gel running buffer at room temperature for 20 min at 90 V (Mini Protean3 power-pack from Bio-Rad, Hercules, CA, USA) and then 120 V until the dye front approached the bottom of the gel. This was conducted in triplicate. Gels were stained with colloidal coomassie brilliant blue G250 (34% methanol (VWR Chemicals, Tingalpa, QLD, Australia), 3% orthophosphoric acid (Merck, Darmstadt, Germany), 170 g/L ammonium sulphate (Bio-Rad, Hercules, CA, USA), 1 g/L coomassie blue G250 (Bio-Rad, Hercules, CA, USA), and de-stained in dH_2_O. 

#### 3.4.4. Thrombin Generation–Calibrated Automated Thrombography (CAT)

Thrombin generation was measured by a Calibrated Automated Thrombography using Thrombinoscope™ software (Thrombinoscope BV, Netherlands) and a Thermo Fisher Fluoroskan fluorometer (Thermo Labsystem, Helsinki, Finland). Eighty microlitres of human plasma (warmed to 37 °C) was manually loaded into a 96-well plate (Thermo Fisher Scientific, Waltham, MA, USA) along with 10 μL of venom in buffer (150 mM NaCl and 50 mM Tris-HCL, pH 7.4) at three concentrations (100, 10, and 1 pg/mL), and 10 μL of phospholipid, in triplicate (final venom concentrations: 830, 83, and 8.3 pg/mL). Calibrator wells contained a kit-provided (cat#86192) thrombin-α2-macroglobulin complex solution of a known concentration, the signal of which all other wells were calibrated against to adjust for internal filter effects, plasma variability, and background activity. Negative controls consisted of venom, vehicle solution, and phospholipid, and of vehicle solution, thrombin calibrator, and phospholipid. All conditions were loaded in triplicate. The plate was placed into the fluorometer and 20 μL fluorogenic substrate in 5 mmol/L CaCl_2_ (FluCa-Kit, cat#86197; Diagnostica Stago) was automatically dispensed, the reaction was run at 37 °C, and fluorescence was read every 20 s for 30 min. Data were automatically converted, calculated, and graphed post-run by Thrombinoscope software. 

#### 3.4.5. Factor X and Prothrombin Activation

Activation of human Factor X (FX) and prothrombin by the venoms were measured with a Fluoroskan Ascent™ Microplate Fluorometer with Ascent^®^ Software v2.6 (Thermo Fisher Scientific, Vantaa, Finland) using the following protocol: 10 µL venom (0.01, 0.05, and 0.1 µg/μL) in dilution buffer (150 mM NaCl, 50 mM Tris-HCl, pH 7.4) was plated in at least triplicate on a 384-well plate along with 10 µL phospholipid (Stago Cat#00597 in O.K. Buffer) and 10 µL of human recombinant FX or prothrombin (0.01 μg/mL) (Haematologic Technologies, VT, USA) in dilution buffer. Venom controls substituted 10 μL coagulation factor with 10 μL dilution buffer. Vehicle controls substituted 10 μL venom and 10 μL coagulation factor with 20 μL dilution buffer. Positive controls substituted 10 μL venom with dilution buffer and 10 μL coagulation factor with 10 µL (0.01 μg/mL) of human recombinant FXa or thrombin (Haematologic Technologies, Inc.). Plates were loaded into the fluorometer (pre-incubated to 37 °C), and 70 µL quenched substrate was dispensed into each well (1 µL substrate in 500 μL enzyme buffer: 150 mM NaCl, 50 mM Tris-HCl, 5 mM CaCl, pH 7.4. Fluorogenic Peptide Substrate Cat#ES002 used for FX activation assay; Cat#ES011 for prothrombin). Fluorescence was monitored (excitation/emission: FX–390 nm/460 nm, prothrombin–390 nm/460 nm) for 300 cycles over ~150 min. 

### 3.5. Statistics

Data were graphed and analysed using GraphPad Prism 9.1.2 (GraphPad Software, San Diego, CA, USA). Data were tested for normality using Shapiro–Wilk and Q-Q plot diagnostic tests, as appropriate. Data are reported as either mean ± SEM or mean with 95% confidence intervals (CI), unless otherwise stated.

Dose-response plasma coagulation curve concentrations were log-transformed, and curve data were normalised between maximal response of the venom’s coagulant activity (i.e., minimum clot time) (100%) and no response (i.e., plasma vehicle control clotting time) (0%). EC_50_s were then calculated using the ‘non-linear regression (log)dose vs (normalised)response–variable slope’ model, unknowns interpolated from standard curve with confidence intervals set to 95%, and extra sum-of-squares F test for comparison between venom best-fit EC_50_s. Mid-curve concentrations were calculated and compared by repeating these steps, except curve data were normalised between clots times induced by 30 μg/mL (100%) and 0.05 μg/mL (0%). Mid-curve clot times were obtained by determining the point at which the mid-curve concentration (x-axis) intersected with the y-axis (clot time). Clotting times of venoms were compared using one-way ANOVA with Tukey’s post hoc analysis, and venoms were compared to controls using unpaired t-tests with Prism default settings. 

For factor activation data, fluorescence emitted by wells containing negative controls and venom controls were subtracted from fluorescence emitted by wells containing venom + zymogen. Due to the extremely high activity of the pure thrombin positive control, the corresponding fluorescence emitted by these wells reached the machine’s maximum (~5000 fluorescence units) before completion of the run and thus the curves plateaued unnaturally. This was rectified by excluding values > 5000 and fitting the thrombin curves using the ‘non-linear curve fit, hyperbola (x is concentration)’ model with 95% confidence intervals to interpolate fluorescence values > 5000. The model predicted the thrombin activity with very high probability (goodness of fit R^2^ > 0.99) and the curve-fit data were thus included in subsequent calculations. Activity for each venom and positive control was calculated by running Area Under the Curve using PRISM default settings. Rate and lag time were obtained by running simple linear regression on the linear portion of curves using PRISM default settings to calculate best-fit slope (rate) and X-intercept (lag time) values with 95% confidence intervals. Data were normalised between zero (0%) and the corresponding positive control (100%) before graphing. 

Thromboelastography data were compared using one-way ANOVA with default PRISM settings or non-parametric equivalents to compare between venoms, and unpaired t-tests or non-parametric equivalents to compare venoms to controls. 

## 4. Conclusions

In the context of these models, *D. siamensis* venoms exhibited the most potent activities within the genus, and, of these, the venom from Myanmar was consistently amongst the strongest, while the Sri Lankan *D. russelii* venom was generally the weakest of the Asian *Daboia* clade. Of the Afro-Arabian clade, only *D. mauritanica* possessed procoagulant venom, while *D. deserti* and *D. palaestinae* venoms induced an anticoagulant effect in plasma. However, both anticoagulant and procoagulant venom properties were mutually evident within most venoms, as all but *D. siamensis* from Myanmar demonstrated anticoagulant fibrinogenolysis, though by widely differing mechanisms, and all acted upon factor X and prothrombin in a procoagulant manner. Synergistic toxin activity is well-documented [65], and as clotting function is not severely compromised until blood factors drop below 10% of their typical plasma concentrations [66], venoms that consume multiple factors in a manner that renders them unavailable for haemostatic regulation—irrespective of whether this is via activation or inhibition—may expedite the induction of haemorrhage and cardiovascular collapse. However, this also exemplifies a pervasive problem in exploring adaptive value of venom function, whereby using a narrow scope to infer the actions of a complex trait within a highly complex biological system can lead to overinterpretation. Epiphenomenal or “promiscuous” protease activity is likely ubiquitous within venoms [52], and both synergistic and competitive actions of toxins occur in parallel. While a given activity may be observed in vitro, this is driven by the choice of lens being used and does not necessarily translate to such actions within a biological system. This represents an understudied research area, and additionally highlights the complexity of biological systems interactions. 

While evolutionary interpretations are speculative, these diverse actions on blood factors are nonetheless both intriguing and clinically relevant. Variation in venom activity of *D. palaestinae* was evident across most assays in this study, as were significant differences between the *D. russelii* and *D. siamensis* variants. Should the asserted *D. deserti*-*mauritanica* species reclassification be widely adopted, significant differences also evident between their venoms indicates that intraspecific venom variation is characteristic of all *Daboia* species. Recognising these differences between venoms is of critical importance for future antivenom production and marketing strategies. 

## Figures and Tables

**Figure 1 ijms-22-13486-f001:**
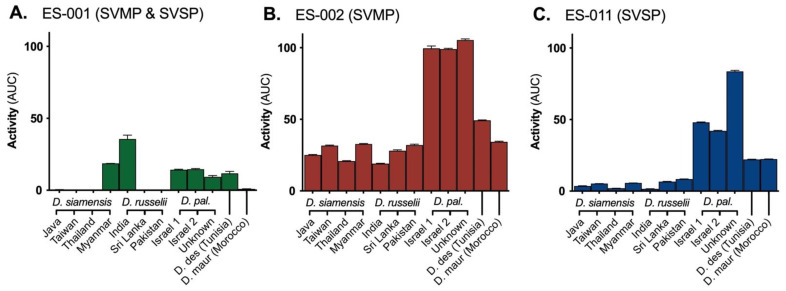
Activity (represented as area under the curve (AUC: relative fluorescence units x time)) of 5 ng/mL *Daboia* venoms on fluorogenic substrates for (**A**) matrix metalloproteases and serine proteases (substrate ES-001), (**B**) matrix metalloproteases (ES-002), and (**C**) serine proteases (ES-011), used to infer relative activity of snake venom serine proteases (SVSP) and snake venom metalloproteases (SVMP). Data are the mean with 95% confidence intervals of three replicates; D. pal = *D. palaestinae*, D. des = *D. deserti*, D. maur = *D. mauritanica*.

**Figure 2 ijms-22-13486-f002:**
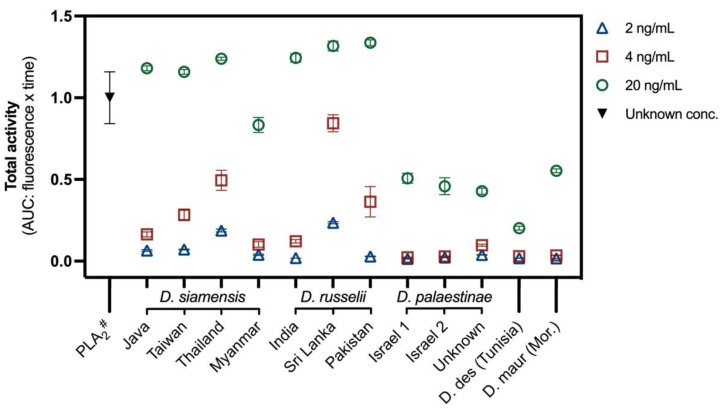
Enzymatic activity (represented as area under the curve (AUC: relative fluorescence units x time)) of 2, 4, and 20 ng/mL µg of *Daboia* venoms on phospholipase A_2_ (PLA_2_)-specific fluorogenic substrate. Data are the mean with 95% confidence intervals of three replicates, normalised to the PLA_2_ control (# kit-provided PLA_2_ (from bee venom) of unknown concentration); D. des = *D. deserti*, D. maur = *D. mauritanica*, Mor. = Morocco.

**Figure 3 ijms-22-13486-f003:**
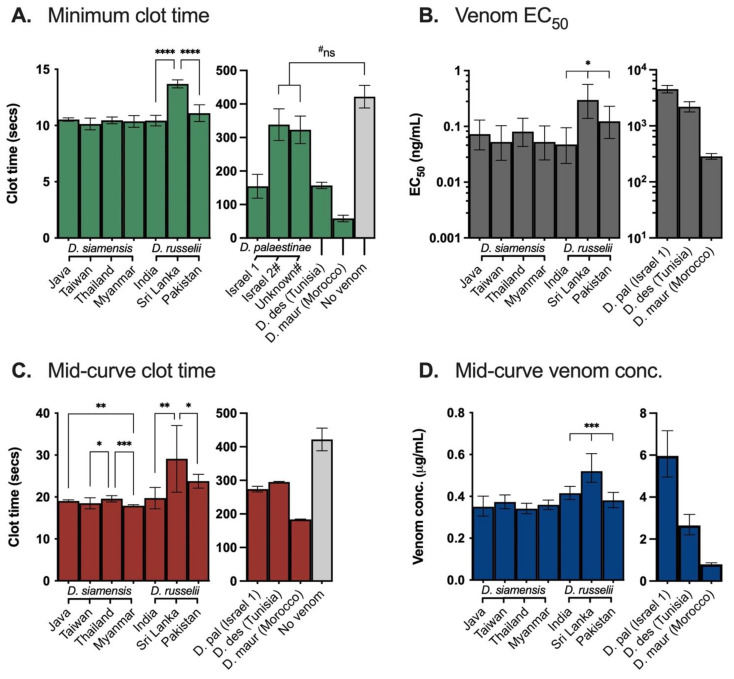
Clotting parameters of procoagulant activity of *Daboia* venoms incubated with human plasma: (**A**) Minimum clot time of plasma at venom saturation point (20 µg/mL for Asian clade (**left**), 30 μg/mL for Afro-Arabian clade (**right**)). (**B**) Venom concentrations required to induce half the maximal response (EC_50_) of procoagulant activity. Data are extrapolated from 9-point dose-response curves (0.05–30 µg/mL) using non-linear regression, normalised between maximal response (minimum clot time) and no response (negative control). Lower EC_50_s represent greater venom potency. (**C**) Mid-curve clot times by venoms tested at nine concentrations between 0.05–30 µg/mL. (**D**) Venom concentrations that induce mid-curve clot times (panel **C**), calculated via non-linear regression and normalised between clotting response at 30 μg/mL and at 0.05 μg/mL. Lower values represent greater venom potency. (**A**–**D**) Venom data are the mean (**A**,**C**) or best-fit values (**B**,**D**) with 95% confidence intervals of at least three replicates; negative control (clot time with no venom) is cumulative mean ± SD of negative controls (50 replicates) conducted over multiple coagulation assays; statistics displayed are the results of one-way ANOVA and Tukey’s tests of intra-species comparisons (i.e., comparisons of multiple venom samples from the same species); **** = *p* < 0.0001, *** = *p* < 0.001, ** = *p* < 0.01, * = *p* < 0.05; D. pal = *D palaestinae*, D. des = *D. deserti*, D. maur = *D. mauritanica*. # Venoms were not procoagulant (ns = not significant (*p* > 0.05); unpaired t-tests of venom vs no venom clot times) and were thus excluded from analyses (**B**–**D**).

**Figure 4 ijms-22-13486-f004:**
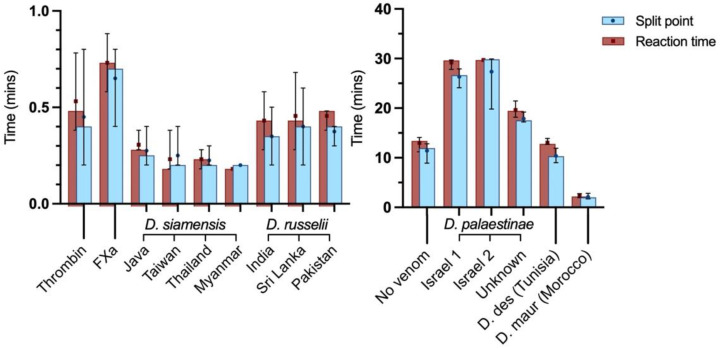
Thromboelastography of human plasma incubated with 20 μg/mL venoms of: **Left**) seven populations of the Asian clade of *Daboia*; **Right**) five populations of the Afro-Arabian clade *Daboia*. A total of 20 μL/mL of thrombin and FXa were used as positive controls. Split point (SP), the time at which clot formation begins, signifies the initiation period of the enzymatic activity. Reaction time (R) denotes the time at which the clot has reached an amplitude of 2 mm. R–SP = conversion rate of prothrombin into thrombin, whereby a greater difference indicates a slower rate. The assay runs for 30 min at 37 °C. Data presented are the mean (symbols), median (bars), and range (error bars) of four independent replicates. D. des = *D. deserti*, D. maur = *D. mauritanica*.

**Figure 5 ijms-22-13486-f005:**
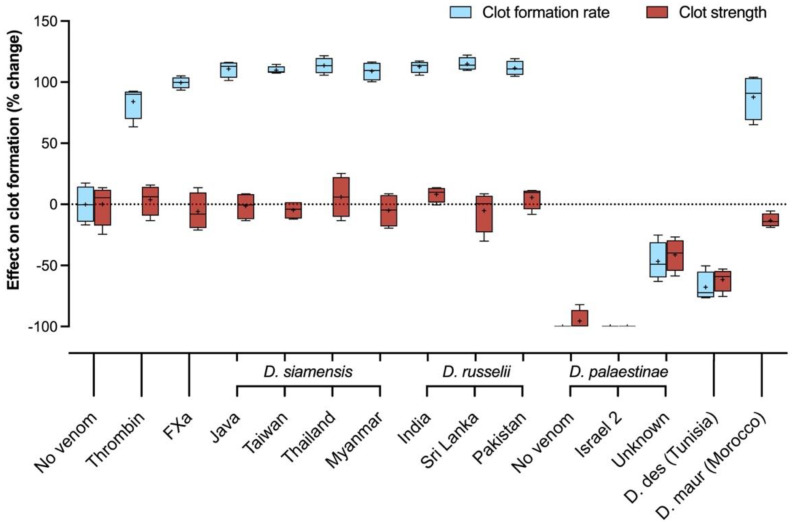
Thromboelastography of human plasma incubated with 20 μg/mL venoms of *Daboia*. A total of 20 uL/mL of thrombin and FXa were used as positive controls. Angle (degrees; pale blue, left y-axis) represents the rapidity of clot formation as a combined measure of enzymatic activity and fibrinogen function; maximum amplitude (mm; dark red, right y-axis) represents clot strength, where higher amplitude denotes greater strength. The assay runs for 30 min at 37 °C. Box plots depict median (line), mean (+), and range (error bars) of four independent replicates; D. des = *D. deserti*, D. maur = *D. mauritanica*.

**Figure 6 ijms-22-13486-f006:**
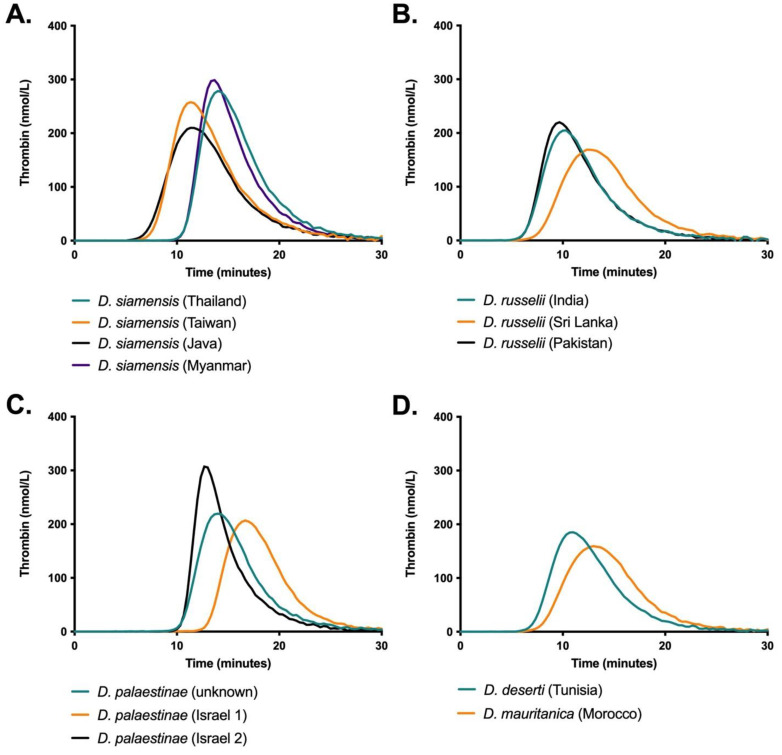
Real-time thrombin generation (nmol/L) by *Daboia* venoms (83 pg/mL) in human plasma, incubated at 37 °C for 30 min, measured by Calibrated Automated Thrombinoscope (CAT). Data are the mean of three replicates (traces are the automatic output of CAT software in which data are automatically calculated and errors bars omitted).

**Figure 7 ijms-22-13486-f007:**
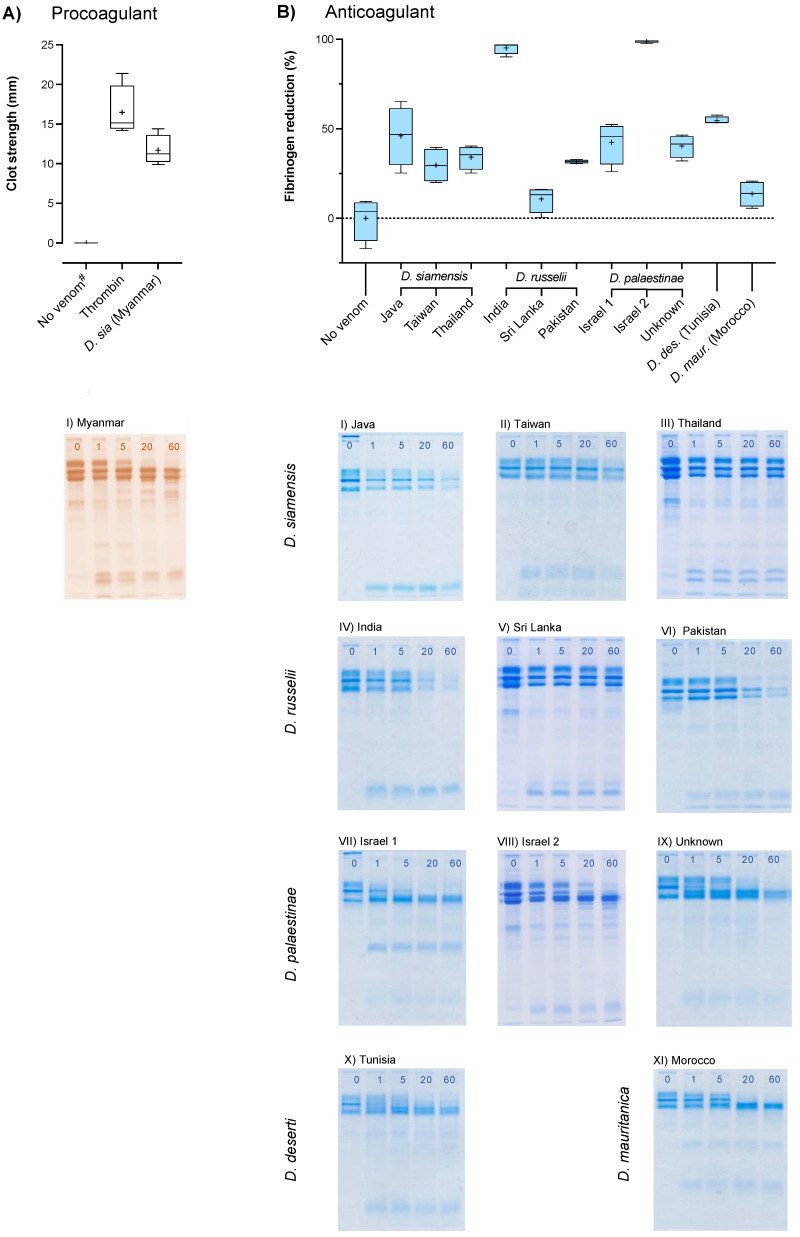
Procoagulant (column **A**) and anticoagulant (column **B**) activities of twelve *Daboia* venoms on human fibrinogen over time measured by thromboelastography (0.1 mg venom per 10 mg fibrinogen per final mL) (**top row**) and fibrinogen gels (0.1 mg/1 mg/mL) (**bottom row**), incubated at 37 °C for up to 60 min. Thromboelastography conditions were: fibrinogen and venom (procoagulant test conditions), fibrinogen and thrombin (procoag. pos. cont.), and fibrinogen and vehicle solution (procoag. neg. cont.) combined and measured for 30 min (figure **A**), after which thrombin was added to any samples that had not formed clots so that: [fib. and venom for 30 min] + thrombin (anticoag. test cond.), and [fib. + vehicle for 30 min] + thrombin (anticoag. neg. cont.) combined and measured for 30 additional min (figure **B**). Box plots depict median (line), mean (+), and range (error bars) of four independent replicates of: (**A**) Clot sizes by venoms that stimulated clotting in the procoagulant test conditions, # = no spontaneous clotting occurred; and (**B**) Reduction in functional fibrinogen (%) by all other venoms assessed in the anticoagulant test conditions, normalised between the anticoagulant negative control clot size (0% reduction) and zero (100% reduction). Fibrinogen gel images are one representative from three independent replicates; lane 1 = no venom and 0 min incubation, lanes 2–5 = 1, 5, 20, 60 min incubation; panels AI-BIII = *D. siamensis*, BIV-VI = *D. russelii*, BVII-IX = *D. palaestinae*, BX = *D. deserti*, BXI = *D. mauritanica*.

**Figure 8 ijms-22-13486-f008:**
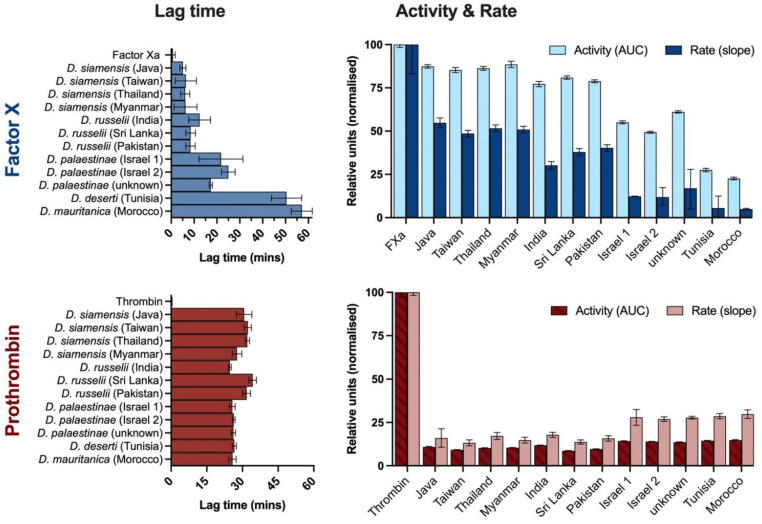
Lag time (minutes until commencement of linear phase; **left column**), total activity (area under the curve (AUC) of fluorescence over time; right column) and peak rate (slope at linear phase; **right column**) of 100 ng/mL of human factor X (**top row**, blue) and prothrombin (**bottom row**, red) cleavage by 100 ng/mL *Daboia* venoms, incubated at 37 °C for 150 min. Activation is calculated as the relative luminescence units (RLU) emitted by wells containing venom + zymogen + substrate after subtraction of venom + substrate control RLU. Data are the mean with 95% confidence intervals of at least three replicates; activity and rate data are normalised to the positive control (100 ng/mL human factor Xa (FXa) or thrombin).

**Table 1 ijms-22-13486-t001:** Details of the venoms used for the study. All venoms were from adult specimens.

Venom	Locality	Known Details
*D. deserti*	Tunisia	Commercially sourced from Latoxan, France. Pooled from captive individuals.
*D. mauritanica*	Morocco	Commercially sourced from Latoxan, France. Pooled from captive individuals.
*D. palaestinae*	Israel	Obtained from a captive individual in Europe by authors BGF and FJV.
*D. palaestinae*	Israel	Obtained from a captive individual in Europe by authors BGF and FJV.
*D. palaestinae*	Unknown	Obtained from a captive individual in Europe by authors BGF and FJV.
*D. russelii*	India	Historical venom sample collected from a single individual that arrived in the United Kingdom (UK) on a boat shipment from Mumbai, India in 1998, and maintained in the Liverpool School of Tropical Medicine (LSTM) herpetarium until 2002
*D. russelii*	Sri Lanka	Commercially sourced from Latoxan, France. Pooled from captive individuals.
*D. russelii*	Pakistan	Thatta District (24°44′46″ N 67°55′28″ E) Sindh province, Pakistan by SAA.
*D. siamensis*	Myanmar	Commercially sourced from Latoxan, France. Pooled from captive individuals.
*D. siamensis*	Thailand	Commercially sourced from Latoxan, France. Pooled from captive individuals.
*D. siamensis*	Taiwan	Commercially sourced from Latoxan, France. Pooled from captive individuals.
*D. siamensis*	Java	Obtained from a captive individual in Europe by authors BGF and FJV.

## Data Availability

All data is included in manuscript figures.

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
