# Peer review of "A Genus-Wide Bioactivity Analysis of Daboia (Viperinae: Viperidae) Viper Venoms Reveals Widespread Variation in Haemotoxic Properties"

_ijms, 2021, doi:10.3390/ijms222413486_

Round 1

Reviewer 1 Report

The submitted article titled - A genus-wide bioactivity analysis of Daboia (Viperinae: Viperidae) viper venoms reveals widespread variation in haemotoxic properties – gives the impression to be in general nicely written, but needs major improvement:

Please follow the general order of article sections:

Introduction – Methods and Materials – Results – Discussion

Please establish the awaited section order!

Abstract please update accordingly – based on the corrections made in the following sections: Introduction – Methods and Materials – Results – Discussion or is this the paper style??

Introduction

please add a map showing the distribution of the different Daboia species

Please specify the used denomination for Viperidae and Viperinae e.g. Oppel, 1811

Please improve readability in lines 81ff - Snake Venom MetalloProteases (SVMPs) and introduce the used abbreviations e.g. PLA2s

Please add the correct devices and technologies – e.g. TEG 5000 and fluorometry, which is in my opinion commonly addressed as the chromogenic method (used in hemostaseology) – please specify.

Material and Methods:

Table 1 please introduce abbreviations and insert links to the companies

This link does not give you a scientific impression!

https://geohack.toolforge.org/geohack.php?pagename=Thatta&params=24_44_46_N_67_55_28_E_type:city_region:PK

Please improve!

Plasma – please specify the general coagulation results

46-48 seconds in absence of a coagulation activator is impossible! Please correct the presented numbers, were to find the section coagulation analyses for protocol

Why is SEM used – this seems to be not the correct statistical presentation – please remove this form of statistical polishing. Please describe the used N or n!

Please verify the used coagulation protocols and add the discrepancies to the standard routine protocols and update references.

It would be a scientific wonder if citrated human plasma coagulates without recalcification!

Please correct these obvious errors.

Please describe the coagulation protocols they seem to be used not according to the published protocols! The TEG protocol is interesting – wait and see for 30 minutes than add thrombin – the most potential coagulation activator and you get what you want ?!?!

Please describe the final recalcification status compared to standard coagulation protocols as high doses of CaCL2 are used. Please explain!

Results:

Please harmonize the graphical presentation (according to figure one, stick to the colours) as e.g. two total different presentations are used in figure 1, 2, 6 and so on –Please harmonize!

Line 281 range 0,20-0,20 mins – please describe

Please reduce text, add further graphical results, and harmonize the graphical presentation using Standard deviation(!)

Discussion and Conclusions

Please address and edit as suggested priviously

Supplementary Materials:

Supporting data and supplementary materials are available online at 766 10.6084/m9.figshare.16545441.

Link does not work and material could not be reviewed! Please show in the next round of the review process

References

Please update only 6 out of 66 references are linked, papers of op den Brouw, Vonk, Casewell and Fry are cited

Please add technical/protocol re

Reviewer 2 Report

Lines 88 to 97 should be removed, as the authors do not need to specify the methods in this section.
There is an error on line 106 as the species D.siamensis from Taiwan is cited instead of Myanmar.
In Figure 1, the Y-axis should be% fluorescence units/min, not the area under the curve. In the case of Figure 2, the Y-axis would be relative fluorescence units/min. The activity of what species is taken as a reference?
I don't understand how the statistic is represented in Figure 3. When it is statistically significant, it is not clear what the study groups are.
The results should be separated from the discussion.
It would be easier to read if the authors incorporate tables summarizing the results in the sections "Intensity of clotting activity" and "Strength of clotting activity".
Figure 7 could become supplementary material and a table could be added here.
The article is excessively long and difficult to follow. The results should be summarized in tables or graphs. The conclusions are repetitive. A single paragraph summarizing the main conclusions of the paper should be sufficient.

Reviewer 3 Report

The authors discuss the toxins of the genus Daboia viper by comparing different activities by species and habitat. We use a lot of experimental data to show that there are various differences in the species inhabiting the capital Africa region, which inhabits the Asian region. Reviewers feel that sufficient data proves this. However, the data shown in FIGS. 2 and 7 give the impression that it is difficult to understand, so please devise an expression.

Round 2

Reviewer 1 Report

The submitted paper was adopted, but there as still some topics to be addressed:

In the point-to-point-answers lines numbered > 2000  are mentioned, but in the submitted version lines are only available up to 1081.

So I can not follow your line numbering system, sorry, please clarify!

e.g. I can not find line 2073-2074, please specify.

Snake Distribution Map – in my opionion you should at least add a good link to a distribution map, as you do not want to include a good picture/map.

Please follow the scientific standards citing devices and reagents used – this means please include the company, place and country in brakets.

Where did you buy/order the bee venom of unknown PLA2 concentration?

Please add the source as a sort of scientific minimum!

Now the authors specify, that they use coagulation assays of an APTT assay type – please specify this also in your manuscript, as a APTT will not coagluate without the presence of calcium.

Where is line 2074 – please specify the used assay  - it seems to be a APTT Test – this is  a very important information for the reader confident in hemostasis and not just a clot formation time as this is a specific paremeter used in thrombelastography.

General coagulation results  - you are using e.g. APTT reagents – what are the plasma results without venom? You will not induce coagulation without adding CaCl2 and an activator!

An please mark those venoms, which act without calcium!

So please clarify your methodology (with and without calcium) throughout your manuscript – why was this source not cited?

Kini, R. M. (2005). The intriguing world of prothrombin activators from snake venom. Toxicon, 45(8), 1133-1145. https://doi.org/10.1016/j.toxicon.2005.02.019 

It is interesting that the authors answer to the specific questions concernign coagulation assays with a KILLER PHRASE – this is not a comparison between apple and pears – so please clarify!

Concerning the coordinates  - I am wondering using a online tool called GeoHack – very assuring!

Point 17 please include this answer in your manuscript – because the average reader will otherwise not understand your approach!

Concerning the references –  approx. 9 % self – citation is good scientific practice?

Author Response

The submitted paper was adopted, but there as still some topics to be addressed:

In the point-to-point-answers lines numbered > 2000  are mentioned, but in the submitted version lines are only available up to 1081.

So I can not follow your line numbering system, sorry, please clarify!

e.g. I can not find line 2073-2074, please specify.

This has to do with the numbers re-setting once track changes were accepted. In any case, all replies were with sufficient information to be assessable stand-alone without having to cross-check.

Snake Distribution Map – in my opionion you should at least add a good link to a distribution map, as you do not want to include a good picture/map.

As weblinks are not permanent features, we respectfully decline. If people are truly interested, looking it up themselves is not an onerous task.

Please follow the scientific standards citing devices and reagents used – this means please include the company, place and country in brakets.

We have annotated all devices and reagents with the required information already.

Where did you buy/order the bee venom of unknown PLA2 concentration?

Please add the source as a sort of scientific minimum!

Also aready detailed in the methods, this control venom was part of the EnzChek® Phospholipase A2 Assay Kit.

Now the authors specify, that they use coagulation assays of an APTT assay type – please specify this also in your manuscript, as a APTT will not coagluate without the presence of calcium.

Where is line 2074 – please specify the used assay  - it seems to be a APTT Test – this is  a very important information for the reader confident in hemostasis and not just a clot formation time as this is a specific paremeter used in thrombelastography.

As thoroughly and patiently explained in the previous reply to this same reviewer, we did not in fact specify we are using an APTT assay. In fact, we are not. APTT is a completely different assay used for a totally different purpose. Our coagulation tests are entirely different as detailed in the methods. We direct the reviewer to the prior reply as we do not feel it is necessary to repeat ourselves with identical information.

General coagulation results  - you are using e.g. APTT reagents

As thoroughly and patiently explained in the previous reply to this same reviewer, we did not in fact specify we are using an APTT assay. In fact, we are not. APTT is a completely different assay used for a totally different purpose. Our coagulation tests are entirely different as detailed in the methods. We direct the reviewer to the prior reply as we do not feel it is necessary to repeat ourselves with identical information.

– what are the plasma results without venom? You will not induce coagulation without adding CaCl2 and an activator!

An please mark those venoms, which act without calcium!

So please clarify your methodology (with and without calcium) throughout your manuscript –

The venoms in this study do not act without calcium, which is the entire point. As we have detailed already in the paper and patiently explained in the previous response to this same reviewer. We direct the reviewer to the prior reply as we do not feel it is necessary to repeat ourselves with identical information.

why was this source not cited?

Kini, R. M. (2005). The intriguing world of prothrombin activators from snake venom. Toxicon, 45(8), 1133-1145. https://doi.org/10.1016/j.toxicon.2005.02.019 

Because we cited a more recent paper by him.

  1. Kini, R.M.; Koh, C.Y. Metalloproteases affecting blood coagulation, fibrinolysis and platelet aggregation from snake venoms: Definition and nomenclature of interaction sites. Toxins 2016, 8, 284.

It is interesting that the authors answer to the specific questions concernign coagulation assays with a KILLER PHRASE – this is not a comparison between apple and pears – so please clarify!

There is nothing to clarify. Our previous replies were clear.

Concerning the coordinates  - I am wondering using a online tool called GeoHack – very assuring!

Point 17 please include this answer in your manuscript – because the average reader will otherwise not understand your approach!

We feel the methods are clear as they are written and respectfully decline to be wordier.

Concerning the references –  approx. 9 % self – citation is good scientific practice?

Considering we have pioneered the methods being used, yes it is good scientific practice! As we already replied to previously to similar comments by the same reviewer. We direct the reviewer to the prior reply as we do not feel it is necessary to repeat ourselves with identical information.

Reviewer 2 Report

The authors have covered all my concerns.

Author Response

There is nothing to respond to.